# Impact of Adjuvant Modalities on Survival in Patients with Advanced Stage Endometrial Carcinoma: A Retrospective Analysis from a Tertiary Medical Center

**DOI:** 10.3390/ijerph16142561

**Published:** 2019-07-18

**Authors:** Yi-Jou Tai, Heng-Cheng Hsu, Ying-Cheng Chiang, Yu-Li Chen, Chi-An Chen, Wen-Fang Cheng

**Affiliations:** 1Department of Obstetrics and Gynecology, College of Medicine, National Taiwan University, Taipei 100, Taiwan; 2Graduate Institute of Clinical Medicine, College of Medicine, National Taiwan University, Taipei 100, Taiwan; 3Department of Obstetrics and Gynecology, National Taiwan University Hospital Hsin-Chu Branch, Hsin-Chu City 300, Taiwan; 4Graduate Institute of Anatomy and Cell Biology, College of Medicine, National Taiwan University, Taipei 100, Taiwan; 5Graduate Institute of Oncology, College of Medicine, National Taiwan University, Taipei 100, Taiwan

**Keywords:** endometrial neoplasms, adenocarcinoma, endometrioid, chemotherapy, adjuvant, radiotherapy

## Abstract

Adjuvant treatment in advanced-stage (stages III /IV) endometrial carcinomas in terms of tumor grades has not yet been explored. We retrospectively analyzed 194 patients with advanced-stage endometrioid endometrial carcinoma who received surgery, followed by adjuvant therapy, at National Taiwan University Hospital between January 1, 2000 and August 31, 2017. Adjuvant therapies included radiation (RT), chemotherapy alone (CT), and combined modality treatment (CMT: radiation and chemotherapy). The prognostic factors were determined from multivariate survival analyses using Cox regression models. Progression-free survival (PFS) and overall survival (OS) times were estimated with the Kaplan–Meier method. The median follow-up was 45.5 months (range: 6.2–207.9). In grade 1/2 endometrioid carcinoma, neither adjuvant CT nor CMT could prolong PFS significantly compared to RT (CT: HR 1.59, 95% CI 0.64–3.97; CMT: HR 2.03, 95% CI 0.72–5.74). Notably, maximal cytoreduction independently improved PFS (HR 0.31, 95% CI 0.10–0.90). No particular adjuvant treatment provided an OS advantage over the others for grade 1/2 endometrioid carcinomas. However, for grade 3 endometrioid carcinoma, CMT showed OS benefits (HR 0.15, 95% CI 0.03–0.89) compared to RT and CT. In conclusion, maximal cytoreduction should be the goal in patients with grade 1/2 advanced-stage endometrioid carcinomas. Based on our results, patients with grade 3 endometrioid carcinomas might benefit from adjuvant CMT.

## 1. Introduction

Endometrial cancer is the most common gynecologic malignancy in developed and Western countries [1]. Moreover, in Taiwan, it is currently the most common gynecologic malignancy, and the age-adjusted incidence of endometrial cancer has increased 3.7-fold over time, from 3.02 to 14.1 cases per 100,000 person-years from 1991 to 2016 [2,3,4]. International Federation of Gynecology and Obstetrics (FIGO) staging reflects the 5-year survival rate, and the majority of patients have favorable outcomes because of early-stage disease at diagnosis [5]. However, patients with advanced disease have grave outcomes, with 5-year survival rates of about 45% and 25%, respectively, for stages III and IV [6,7]. In addition to FIGO stage, histologic types and tumor grades are prognostic factors. Endometrial cancer is divided into type I and type II tumors based on the clinical, histologic, and molecular features [8,9]. Type II endometrial cancer, which comprises serous, clear cell, and high-grade (grade 3) endometrioid histology, exhibits worse outcomes than type I endometrial cancer, which comprises mucinous and grade 1/2 endometrioid histology.

Surgery remains the cornerstone of endometrial cancer treatment, and postoperative therapy is tailored based on risk factors such as patient age, tumor stage, histologic grade, and myometrial invasion depth [10,11]. Adjuvant therapies typically involve pelvic external beam radiation therapy (EBRT) and/or vaginal brachytherapy, chemotherapy (CT), or both in a combined modality therapy (CMT) [12]. Randomized studies have shown that adjuvant CT plays a major role in the treatment of advanced disease, despite the increased toxicity and pelvic relapse associated with CT alone [13,14,15]. Another approach is to combine CT and radiation therapy (RT), with the intent of controlling both systemic and local recurrences [16,17,18]. However, the GOG 258 trial did not reveal any differences in overall survival (OS) or recurrence-free survival rates between patients treated with CMT and those treated with CT alone [19]. In contrast, the PORTEC-3 trial showed that adjuvant CMT could improve failure-free survival better than adjuvant RT alone, despite the lack of impact on OS [20]. These two trials recruited patients in early stages, with high-risk features (e.g., grade 3, deep myometrial invasion, or serous or clear cell histology), and in advanced stages (stages III/IV). Despite the substantially different clinical presentations and prognoses in different types (types I and II) of endometrial carcinoma, the current management strategies remain similar. Therefore, we conducted this retrospective study to address the outcomes of different adjuvant modalities in patients with advanced-stage endometrioid endometrial carcinoma.

## 2. Materials and Methods

### 2.1. Patients

The medical records of women diagnosed with stage III/IV endometrioid endometrial cancer at National Taiwan University Hospital between January 1, 2000 and August 31, 2017 were retrospectively reviewed. The study protocol was approved by the Institutional Review Boards of our institute. The exclusion criteria were non-endometrioid histology or mixed histology types; stage I or II disease; primary surgery or postoperative adjuvant administered at other institutes; CT or RT given before the operation; and surgery or treatments given with a palliative intent. Patients with synchronous primary malignancies were excluded (Figure 1). The histologic tumor grade was determined based on tumor architecture and the nuclear grading system recommended by FIGO [5]. We divided grades into two categories—grade 1/2 and grade 3—according to previous literature [21].

### 2.2. Surgery

Staging surgery included peritoneal washings, hysterectomy, bilateral salpingo-oophorectomy, and either selective or systematic lymphadenectomy (pelvic with or without para-aortic). The para-aortic lymph nodes were sampled or dissected from the aortic bifurcation to the inferior mesenteric artery in patients with elevated serum CA125, myometrial invasion of >50%, extrauterine spread, or pelvic lymph nodes of >1 cm, as identified in a preoperative MRI (according to the KGOG-2014 criteria) [22]. Bulky nodes were removed by dissection whenever possible. An omental biopsy or omentectomy was performed at the discretion of the individual surgeon, and the decision was made based on the extent of disease. 

### 2.3. Radiation Therapy 

For radiation therapy, patients were treated with EBRT, with or without vaginal brachytherapy. The dose of EBRT was 5040 cGy over 6 weeks, 5 days per week, with a daily fraction of 1.8 Gy. Intensity-modulated radiation therapy or volumetric-modulated arc therapy was also delivered. Pelvic radiotherapy targeted the lower common iliac, external iliac, internal iliac vessels, parametria, upper vagina, and para-vaginal tissues. The presacral lymph nodes were irradiated in patients with cervical involvement. In patients with multiple positive pelvic lymph nodes or documented para-aortic lymph node disease, extended-field radiotherapy was considered. The extended field included the pelvic volume and targeted the entire common iliac lymphatic chain and the para-aortic lymph node region. 

EBRT was performed with 10 MV radiation beams, projected through multiple coplanar ports using the Elekta Synergy accelerator (Elekta, Stockholm, Sweden) or the Varian TrueBeam™ Radiotherapy System (Varian, Palo Alto, CA, USA). The treatment position was verified weekly with cone-beam computed tomography (CBCT) X-ray volume imaging. Following pelvic irradiation, high-dose-rate (HDR) brachytherapy was applied via a vaginal cylinder. Brachytherapy doses were 6 Gy per fraction for two fractions, delivered to the vaginal mucosa using the Nucletron or Varian GammaMed HDR Ir-192 remote afterloading technique. The RT group included patients who received concurrent chemoradiotherapy (CCRT). This treatment consisted of cisplatin, administered weekly at a dose of 40 mg/m^2^, for a total of 5 cycles during EBRT.

### 2.4. Chemotherapy

The CT regimen was administered as per physician preference. Treatments included platinum combined with paclitaxel, anthracycline, cyclophosphamide, or ifosfamide. Treatment modifications included a cycle delay or dose reductions for hematologic toxicities.

### 2.5. Combined Modality Therapy

Patients received one of two types of CMT. In the sequential group, patients received RT followed by CT or, conversely, CT followed by RT. In the sandwich group, patients were first treated with 2 or 3 cycles of CT, followed by interval RT; then, they received an additional 3 to 4 cycles of CT.

### 2.6. Statistical Analyses

Statistical analyses were performed using SPSS software ver. 22.0 (IBM, Armonk, NY). Patient characteristics and clinico-pathologic parameters were evaluated using the Chi-square or Mann–Whitney test. Progression-free survival (PFS) was defined as the time between the date of surgery and the date of documented progression of residual disease, recurrent disease, or death. Overall survival (OS) was calculated as the time between the date of surgery and death or the last date of follow-up, whichever came first. Differences in PFS and OS between groups were evaluated with Kaplan–Meier analyses. Univariate and multivariate Cox proportional hazard regression models were constructed to determine the influence of covariates on survival. A *p*-value of <0.05 was considered statistically significant. 

### 2.7. Details of Ethics Approval

This study was approved by the Research Ethics Committee at the National Taiwan University Hospital (201803076RIN). All patient data were fully anonymized before we accessed them; thus, the Research Ethics Committee waived the requirement for informed consent.

## 3. Results

### 3.1. Patient Characteristics and Distribution of Adjuvant Therapies

Our analyses included 194 patients with advanced-stage endometrioid endometrial carcinoma. The clinical characteristics and pathological factors for these 194 patients are summarized in Table 1. The pie chart (Figure 2) indicates that the distribution of adjuvant therapies varied with the cancer stage. RT accounted for over half of the adjuvant therapies delivered in stages IIIA, IIIB, and IIIC1, while more patients in stages IIIC2 and IV were treated with CT.

### 3.2. Patient Demographics and Disease Patterns According to Tumor Grades

Table 2 summarizes the clinico-pathologic features of grade 1/2 and grade 3 tumors. Grade 1/2 tumors had lower incidences of deep myometrial invasion and lymphovascular space invasion (LVSI). Grade 3 tumors had higher rates of node-positive disease and higher percentages of recurrence or disease progression than grade 1/2 tumors. The median progression-free and follow-up times were significantly shorter in grade 3 than in grade 1/2 tumors. 

### 3.3. Risk Factors for Survival Related to Tumor Grades

Disease relapse and/or progression occurred in 73/194 (37.6%) women. Among these, 10 survived without disease, 22 survived with disease, and 41 died at the time of last contact. 

A univariate analysis showed that among patients with grade 1/2 tumors, stage IV disease, depth of myometrial invasion, residual tumor size, and adjuvant RT were significantly associated with PFS (Appendix A). The multivariate analysis showed that residual tumor status was the only independent predictor of PFS (HR: 0.31, 95% CI: 0.10–0.90, *p* = 0.03) in grade 1/2 tumors. In contrast, FIGO stage and adjuvant therapy were not prognostic factors (Table 3). We next evaluated the OS for grade 1/2 tumors. A univariate Cox regression analysis showed that stage IV disease, lymph node metastasis, and residual tumors of <1 cm were also related to OS (Appendix A). A multivariate regression analysis constructed with these three variables showed that no factor was significantly prognostic of OS for patients in grade 1/2 (Table 3). 

A multivariate analysis showed that the PFS of patients in grade 3 was not significantly related to any of the clinical parameters (Table 3). In contrast, the OS of grade 3 was related to the type of adjuvant therapy (Table 3). CMT improved OS with borderline significance (HR: 0.15, 95% CI: 0.03–0.89, *p* = 0.04) compared to RT.

### 3.4. Survival Outcomes

We used Kaplan–Meier curves to analyze survival outcomes stratified by stage and tumor grade (Figure 3). We observed a significant survival difference between stage III and stage IV disease for different tumor grades. Patients in stage III with grade 1/2 tumors had the best PFS and OS profiles. The median PFS and OS values could not be estimated because over 50% of patients were alive at the last time point. Patients with stage IV disease and with grade 3 tumors had significantly worse survival, with median PFS and OS values of 3.8 and 12.3 months, respectively. 

We analyzed the survival of patients with grade 1/2 tumors stratified by adjuvant therapy. Grade 1/2 tumors treated with RT alone were associated with significantly better PFS than those treated with CT or CMT (*p* = 0.02, log-rank test; Figure 4A). However, OS did not differ significantly between the three adjuvant therapy groups (*p* = 0.16, log-rank test; Figure 4B). Notably, more than half the patients in grade 1/2 did not experience disease relapse or progression during the study period, regardless of the adjuvant therapy administered. 

We analyzed the survival of patients in grade 3 stratified by adjuvant therapy. In grade 3, PFS and OS were similar across all types of adjuvant therapy (Figure 5). CMT showed a trend toward improved PFS and OS, but the difference was not significant. The median PFS was 15.4 months in the RT group and 8.2 months in the CT group; the median (range: 6.8–112.3 months) was not reached for 12 patients in CMT group. Among these 12 patients, 8 were free of disease relapse during the study period. 

## 4. Discussion

Adjuvant treatment modalities for advanced-stage endometrial cancers include systemic therapy and/or RT. Uncertainty remains regarding the best regimens(s) for CT and the benefit of adding RT [23]. RT has been shown to reduce local relapses, but its role in preventing tumor spread remains uncertain. Up to 20%–30% of patients treated with CT alone experienced a high rate of pelvic recurrence as the first site of relapse [24,25]. Consequently, Onda proposed adjuvant chemoradiotherapy [26]. Cisplatin, doxorubicin, and paclitaxel are active agents in treating metastatic or recurrent endometrial carcinoma [27,28,29]. Among patients in our cohort with residual disease (postoperative tumor diameter of ≥1 cm) who were treated with adjuvant CT, 35.7% (5/14) showed a complete response, 28.6% (4/14) showed a partial response, and 35.7% (5/14) developed progressive disease. The response to CT in our study was similar to the 25%–57% response rates reported in previous studies [30].

In this study, the 5-year PFS and OS rates were 64.2% and 84.6%, respectively, in the RT group, and 53.1% and 67.3%, respectively, in the CT group. Survival of advanced endometrial cancer was substantially better in the present study than in previous studies. This difference might be due to the exclusion of other unfavorable histology types (serous/clear cell carcinoma) [31,32]. 

In grade 1/2 tumors, the PFS and OS difference between RT and other adjuvant modalities did not reach significance after adjustment. We found that residual tumor status was independently associated with PFS. Several previous studies evaluated the impact of cytoreduction on survival for advanced-stage or disseminated peritoneal lesions of endometrial cancer. They showed that survival was significantly correlated with the presence of residual disease [33,34]. However, the definition of complete cytoreduction varied across studies. 

In grade 3 tumors, CMT was the only predicting factor for improved survival. Reasons why CMT did not show an improved OS might be the favorable nature of grade 1/2 tumors and the low cumulative incidence of events over time in the CMT group. CMT was advocated in the past decade; thus, it was applied more frequently as an adjuvant therapy in the late study period, and consequently, the follow-up time for CMT was relatively shorter (median: 48.1 months) than for RT and CT (medians: 76.4 and 58.3 months, respectively).

The main strength of this study is that we exclusively focused on women with advanced-stage endometrioid endometrial carcinoma who had adjuvant therapies. We provide a direct comparison of three adjuvant strategies: RT, CT, and CMT. An additional strength is the availability of detailed clinical information and the long follow-up time. The present study had limitations because of its retrospective nature, small sample size, and mismatched distribution of three adjuvant therapy types in grade 1/2 and grade 3 groups. Other limitations are the absence of systematic para-aortic lymphadenectomy performance and a bias in the administration of adjuvant therapy; therefore, the results should be interpreted cautiously.

## 5. Conclusions

CMT did not improve the PFS or OS outcome of patients with grades 1/2 endometrioid adenocarcinoma compared to RT. We recommend resecting grossly visible tumors as completely as possible during surgery for patients with grades 1/2 endometrioid adenocarcinomas. For grade 3 tumors, CMT might improve survival. However, further investigation is required to determine the value of different adjuvant therapy modalities for patients with grade 3 tumors.

## Figures and Tables

**Figure 1 ijerph-16-02561-f001:**
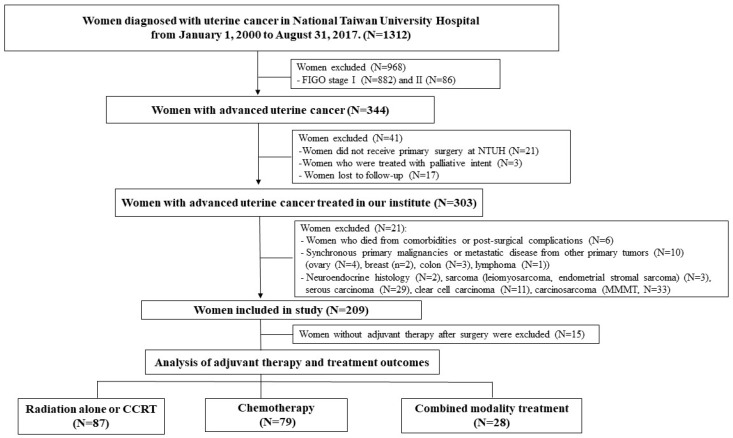
Flow chart of the study population.

**Figure 2 ijerph-16-02561-f002:**
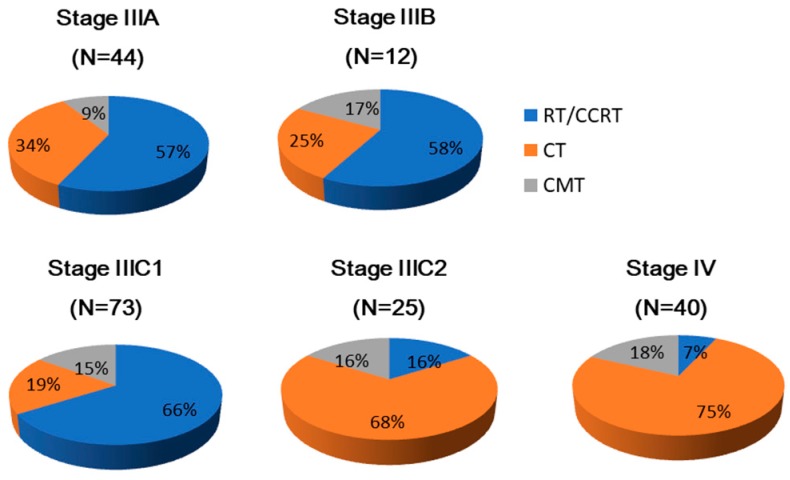
Distribution of adjuvant therapies. The number of patients in each stage is indicated in parentheses. The labels displayed in each chart represent the percentages of patients who received the different adjuvant therapies.

**Figure 3 ijerph-16-02561-f003:**
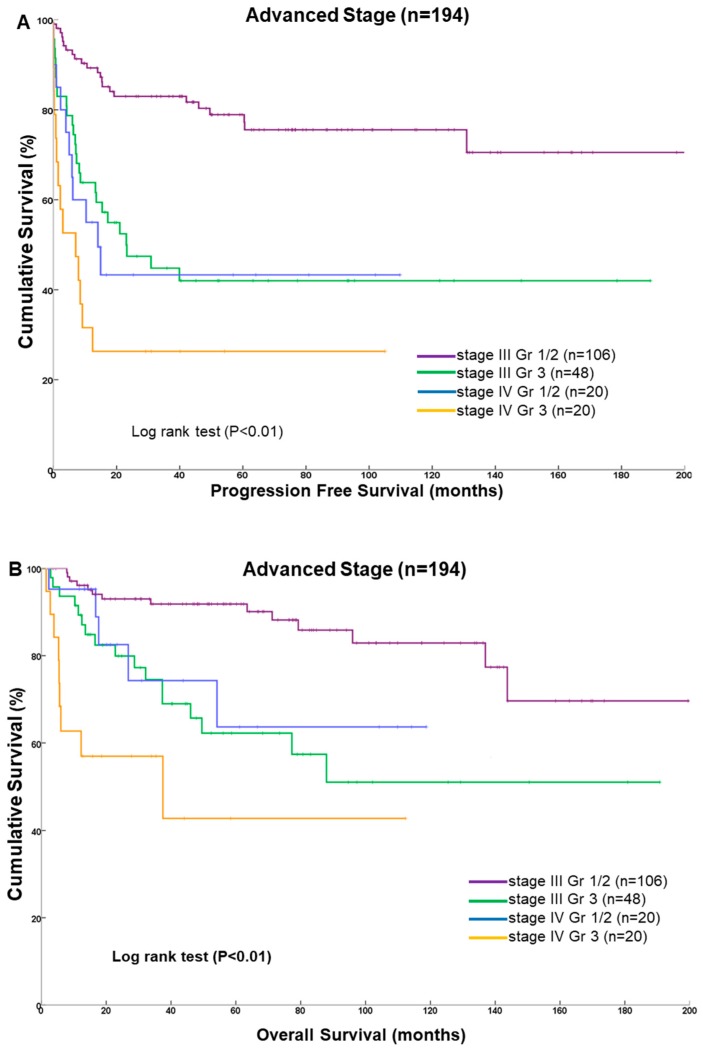
Kaplan–Meier estimates of survival for 194 patients with advanced-stage endometrioid endometrial carcinoma. (**A**) Progression-free survival; (**B**) overall survival. Four distinct subgroups represent different disease stages and tumor grades.

**Figure 4 ijerph-16-02561-f004:**
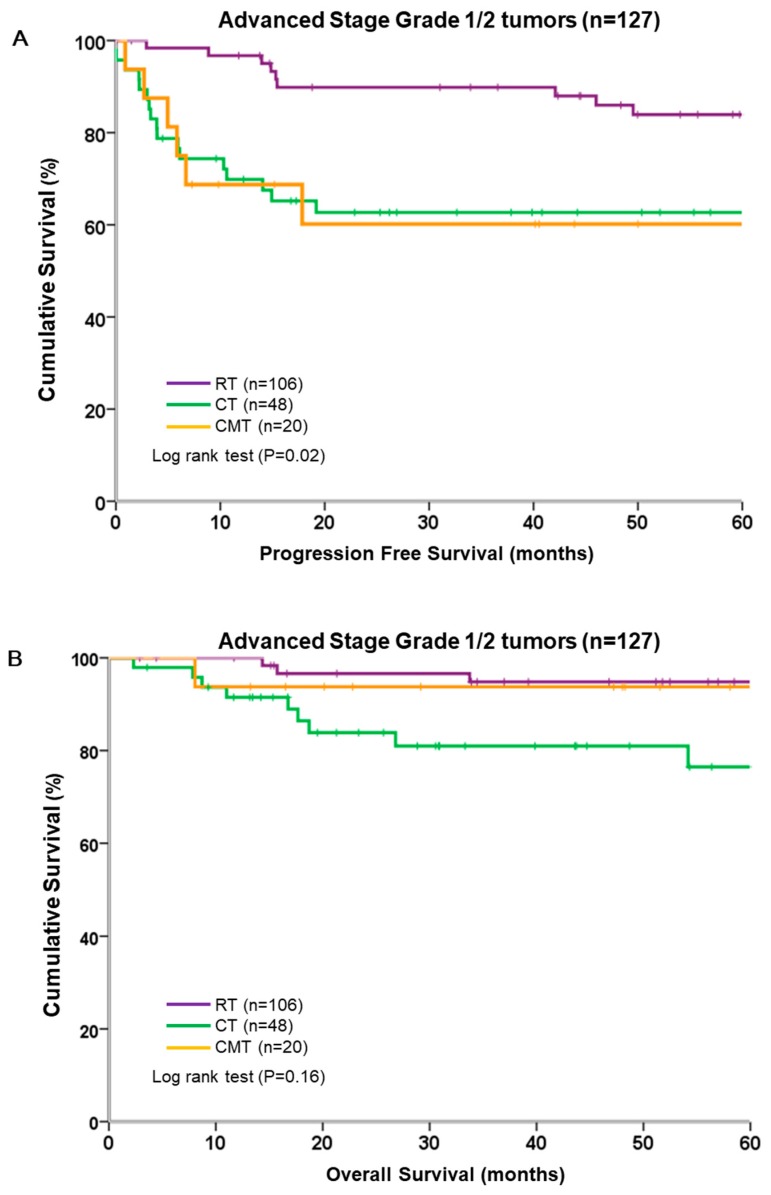
Survival of 127 patients with grade 1/2 tumors, stratified by different adjuvant modalities. (**A**) Progression-free survival; (**B**) overall survival.

**Figure 5 ijerph-16-02561-f005:**
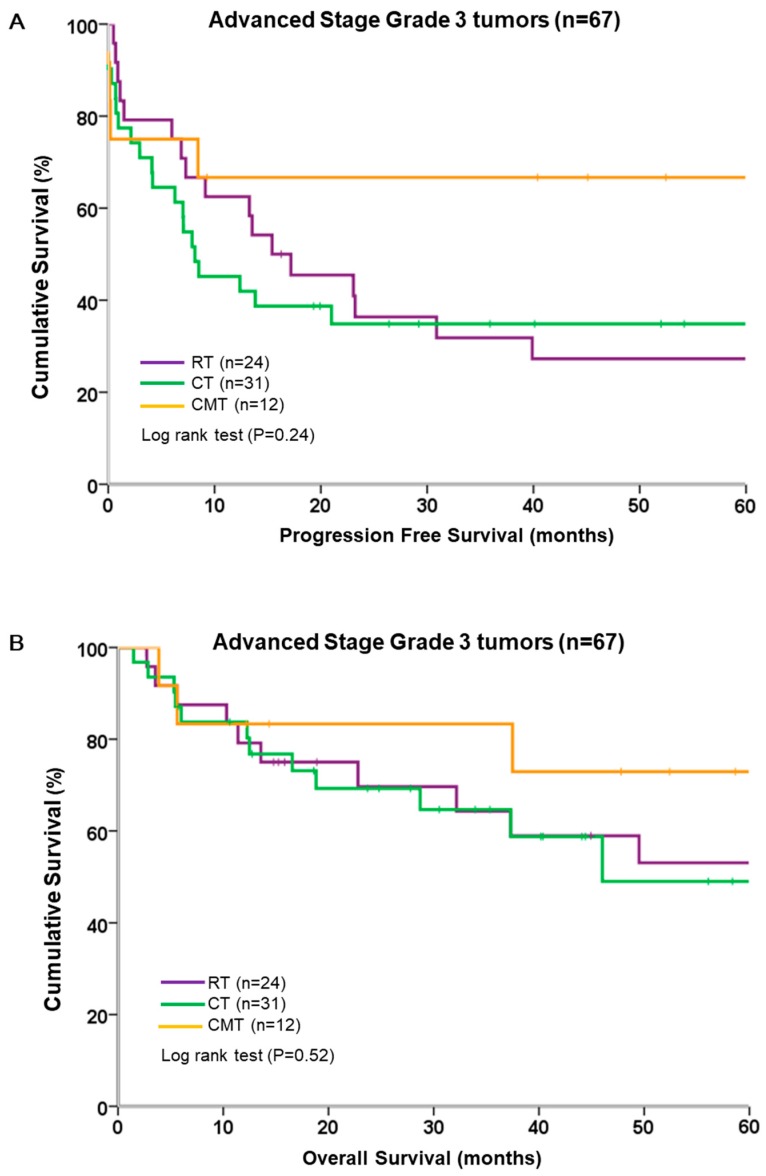
Survival of 67 patients with grade 3 tumors, stratified by different adjuvant modalities. (**A**) Progression-free survival; (**B**) overall survival.

**Table 1 ijerph-16-02561-t001:** Characteristic of patients (*n* = 194) with advanced-stage endometrioid endometrial carcinoma.

Characteristics	Number (%)
Age at diagnosis (years) (median, range)	55.0 (26–81)
Preoperative CA-125 (U/mL) (median, range)	62.7 (3.7–4544.7)
FIGO stage	
IIIA	44 (22.7)
IIIB	12 (6.2)
IIIC1	73 (37.6)
IIIC2	25 (12.9)
IVA	3 (1.5)
IVB	37 (19.1)
Lymph node sampling/dissection (*n* = 184)	
Pelvic alone	132 (68.0)
Pelvic and para-aortic	52 (26.8)
Number of pelvic lymph nodes retrieved (median, range)	16 (1–60)
Number of para-aortic nodes retrieved (median, range)	6 (1–19)
Residual tumor size <1 cm or >1 cm	163 (84.0), 31 (16.0)
Histological grade 1, 2, 3	68 (35.1), 59 (30.4), 67 (34.5)
Lymph node metastases *	
Pelvic or para-aortic nodes alone	97 (52.7), 4 (2.2)
Both pelvic and aortic nodes	18 (9.8)
Postoperative adjuvant therapy	
RT ^†^	87 (44.8)
CT	79 (40.7)
CMT	28 (14.4)
Chemotherapeutic regimens (*n* = 113)	
Platinum and paclitaxel	86 (44.3)
Platinum and anthracyclines	16 (8.2)
Platinum alone	6 (3.1)
Platinum with the other regimens	3 (1.5)
Platinum, anthracycline, and paclitaxel	2 (1.0)
Follow-up time (months) (median, range)	45.5 (6.2–207.9)

* Percentage of patients with involved nodes was counted as the proportion of patients with involved nodes out of all patients who underwent lymph node sampling/dissection. ^†^ The RT group included 87 patients (81 RT alone and 6 CCRT).

**Table 2 ijerph-16-02561-t002:** The clinico-pathologic characteristics of grade 1/2 and grade 3 endometrioid endometrial carcinoma in 194 advanced-stage patients.

Characteristics	Grade 1/2 (*n* = 127)	Grade 3 (*n* = 67)	*p* Value
Age (years) median (range)	54.9 (26–76)	55.4 (29–80.7)	0.60
BMI (kg/m^2^) median (range)	24.4(14.2–42.7)	24.9 (15.2–39.4)	0.80
CA-125 (U/mL)	58.8 (3.7–1376)	79.1 (6.9–4545)	0.72
FIGO stage (2009)			
III IIIA	36 (28.3%)	8 (11.9%)	0.04
IIIB	6 (4.7%)	6 (9.0%)	
IIIC1	50 (39.4%)	23 (34.3%)	
IIIC2	14 (11%)	11 (16.4%)	
IV IVA	2 (1.6%)	1 (1.5%)	0.52
IVB	19(15%)	18 (26.9%)	
Primary tumor size			
<2 cm	22 (17.3%)	5 (7.5%)	0.07
≥2 cm	105 (82.7%)	62 (92.5%)	
Depth of myometrial invasion			
< 1/2	57 (44.9%)	18 (26.9%)	0.04
≥1/2	70 (55.1%)	49 (73.1%)	
LVSI	76 (59.8%)	61 (91 %)	0.001
Lymph node metastases *			
Yes	74 (58.3%)	48 (71.6%)	0.03
No	50 (39.4%)	12 (17.9%)	
Postoperative adjuvant therapy			
RT	63 (49.6%)	24 (35.8%)	0.14
CT	48 (37.8%)	31 (46.3%)	
CMT	16 (12.6%)	12 (17.9%)	
Treatment outcome			
Recurrence and/or progression	32 (25.2%)	41 (61.2%)	<0.01
Death (%)	15 (11.8%)	26 (38.8)	<0.01
Progression-free time (months)	47.2 (14.0–86.5)	10.3 (1.8–40.0)	<0.01
Follow-up (months)	57.5 (22.5–96.9)	26.3 (10.4–57.8)	<0.01

Data are shown as cases (%); CA-125 level, progression-free time, and follow-up time shown as median (25th–75th percentile). LVSI: Lymphatic vascular space invasion. * Lymphadenectomy omitted in 3 patients in the grade 1/2 group and in 7 patients in the grade 3 group.

**Table 3 ijerph-16-02561-t003:** Multivariate survival analysis for grade 1/2 and grade 3 tumors in advanced-stage patients.

	Grade 1/2				Grade 3			
	PFS		OS		PFS		OS	
	HR (95% CI)	*p*	HR (95% CI)	*p*	HR (95% CI)	*p*	HR (95% CI)	*p*
FIGO stage								
IIIA	1		1		1		1	
IIIB	2.89 (0.69,12.2)	0.15	3.65 (0.32,41.6)	0.22	0.80 (0.18,3.60)	0.77	0.18 (0.02,1.78)	0.44
IIIC (IIIC1+IIIC2)	1.12 (0.41,3.08)	0.83	2.70 (0.55,13.2)	0.30	0.38 (0.07,2.15)	0.27	0.42 (0.12,1.48)	0.05
IV (IVA+IVB)	1.32 (0.31,5.65)	0.71	5.75 (0.95,34.7)	0.06	1.37 (0.22,8.62)	0.74	1.49 (0.36,6.10)	0.58
Myometrial invasion								
<1/2	1	0.18	1	0.74		0.48	1	0.49
≥1/2	1.78 (0.77,4.10)		1.19 (0.43,3.30)		1.43 (0.52,3.93)		1.39(0.55,3.51)	
Lymph nodes metastases								
No	1	0.06	1	0.93	1	0.61	1	0.09
Yes	2.75 (0.96,7.90)		1.23 (2.15,9.24)		1.45 (0.35,6.11)		9.46(0.71,125.4)	
Residual tumor size								
≥1 cm	1	0.03	1	0.25	1	0.45	1	0.71
<1cm	0.31 (0.10,0.90)		0.36 (0.06,2.09)		0.76 (0.25,1.81)		1.24 (0.14,2.31)	
Adjuvant therapy								
RT	1		1		1		1	
CT	1.59 (0.64,3.97)	0.32	1.54 (0.43,5.51)	0.51	0.91 (0.40,2.06)	0.82	0.40 (0.11,1.38)	0.15
CMT	2.03 (0.72,5.74)	0.18	0.42 (0.05,3.93)	0.45	0.31 (0.09,1.11)	0.07	0.15 (0.03,0.89)	0.04

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
