# Peer review of "Impact of Adjuvant Modalities on Survival in Patients with Advanced Stage Endometrial Carcinoma: A Retrospective Analysis from a Tertiary Medical Center"

_ijerph, 2019, doi:10.3390/ijerph16142561_

Round 1
Reviewer 1 Report
Impact of Adjuvant Modalities on Survival in Patients with Advanced Stage Endometrial Carcinoma: A retrospective analysis from a tertiary medical center
Authors compare different adjuvant treatment effects in advanced-stage endometrial carcinomas. Authors analyze the progression-free survival (PFS) and overall survival (OS) using Cox regression models for those patients with adjuvant treatment. For the results, authors find that PFS can be prolonged for grade ½ endometrioid carcinoma with RT and OS can be found to be prolonged for grade 3 endometrioid carcinoma with CMT. However, there are some major concerns in the manuscript.
1. In the Introduction, authors should state the relationship and/or difference of the terms used in the manuscript, such as tumor grade ½ and grade 3, and stage III and stage IV, which can help understand the manuscript.
2. Line 155, the conclusion of “The median PFS and OS times were significantly shorter in grade 3 than in grade 1/2 tumors” can be got from Table 3 but not from Table2. Authors should make a clear description in the manuscript.
3. Line 164, authors used univariate analysis to analyze the PFS. I think authors can use a step-wise regression model to select the top features with the great variance that affect the PFS.
4. For Figure 5, samples with CMT treatment is only 12. The sample size might be too small to demonstrate the conclusion.
Author Response
1. In the Introduction, authors should state the relationship and/or difference of the terms used in the manuscript, such as tumor grade ½ and grade 3, and stage III and stage IV, which can help understand the manuscript.
Answer: Thank you for your comments and recommendations. We described the prognostic factors such as disease stage and tumor grades (Page 2, Line 51). The stage allocation and tumor grading were based on the FIGO system. The reason for grade 1/2 and grade 3 categorization was described (Material and Method section, Page 2, Line 83).
2. Line 155, the conclusion of “The median PFS and OS times were significantly shorter in grade 3 than in grade 1/2 tumors” can be got from Table 3 but not from Table2. Authors should make a clear description in the manuscript.
Answer:
The median progression-free time was 47.2 and 10.3 months for grade 1/2, grade 3 tumor, respectively. The median follow-up time was 57.5 and 26.3 months grade 1/2, grade 3 tumor, respectively. Mann Whitney test was used to assess differences between groups. We changed the wording to avoid possible confusion (Page 6, Line 155).
3. Line 164, authors used univariate analysis to analyze the PFS. I think authors can use a step-wise regression model to select the top features with the great variance that affect the PFS.
Answer:
We used the stepwise regression model to analyze factors that affect survival.
To analyze factors that affect PFS and OS in grade 1/2 tumors
PFS:
We ran a stepwise regression with 194 patients and 8 variables (age, menopause status, FIGO stage, depth of myometrial invasion, presence of LVSI, lymph node metastasis, residual tumor size and type of adjuvant therapy). We used SPSS stepwise, which are an entry level and stay level of 0.15; in forward, an entry level of 0.50, and in backward a stay level of 0.10. Forward selection yielded a final model with one variable (residual tumor size >1 or<1 cm) significant at p <0.05. 2="" backward="" selection="" yielded="" variables="" presence="" of="" residual="" tumor="" and="" one="" variable="" size="">1 or<1 cm) significant at p <0.05.< span="">
OS:
Forward stepwise model, residual tumor size (< 25px) maintained significant, with an odds ratio of 0.26 (P = 0.02). Backward selection yielded one variables (residual tumor size) but no variable was significant for OS
To analyze factors that affect PFS and OS in grade 3 tumors
PFS:
Using multiple stepwise linear regression analysis, no significant factor increased PFS.
OS:
Backward selection yielded one variables (type of adjuvant therapy) but not significant at level of p<0.05.< span="">
4. For Figure 5, samples with CMT treatment is only 12. The sample size might be too small to demonstrate the conclusion.
Answer: Thank you for your advice. The small number of patients with CMT and retrospective nature of the study design did not allow us to conclude CMT improved survival. We have revised the sentence in Page 1, Line 38 and Page 7, Line 167.
Reviewer 2 Report
This manuscript is a simple but useful report on the outcome of various treatment to the patients with advanced stage endometrial cancer. The analysis is straightforward and appears to be carried out adequately. However, I have a question. It was stated that there are two patient groups depending on the type of CMT that they were received: subsequent group and sandwich group. I wonder if there is any difference in the outcome of the treatment between these two groups.
Apart from that, the full details of every abbreviation would be good.
Author Response
Answer: Thank you for your comment.
A total of 28 patients received CMT (both radiation and chemotherapy) in our study: 10 treated with sequential therapy and 18 treated with sandwich therapy.
Limited number of patients in CMT group raises statistical issue to conclude which regimen (sequential or sandwich) is superior.
Round 2
Reviewer 1 Report
Authors have addressed most of my concerns.
The statistical methods are simple for small sample size. Authors have tried forward stepwise regression method, but fewer features can be selected. It might be improved with an increase of sample size.
Author Response
Thank you for your recommendations. We made changes in our discussion (Page 11, Line 244) and described the limitation of our analysis.